# Impact of HCV Eradication on Lipid Metabolism in HIV/HCV Coinfected Patients: Data from ICONA and HepaICONA Foundation Cohort Study

**DOI:** 10.3390/v13071402

**Published:** 2021-07-19

**Authors:** Martina Spaziante, Gloria Taliani, Giulia Marchetti, Alessandro Tavelli, Miriam Lichtner, Antonella Cingolani, Stefania Cicalini, Elisa Biliotti, Enrico Girardi, Andrea Antinori, Massimo Puoti, Antonella d’Arminio Monforte, Alessandro Cozzi-Lepri

**Affiliations:** 1Clinical Epidemiology Unit, National Institute for Infectious Diseases Lazzaro Spallanzani IRCCS, 00149 Rome, Italy; martina.spaziante@uniroma1.it (M.S.); enrico.girardi@inmi.it (E.G.); 2Department of Translational and Precision Medicine, Sapienza University of Rome, 00185 Rome, Italy; elisa.biliotti@uniroma1.it; 3Task Force Anti-COVID, AORN San Giuseppe Moscati, 83100 Avellino, Italy; 4ASST Santi Paolo e Carlo, Clinic of Infectious and Tropical Diseases, Department of Health Sciences, University of Milan, 20122 Milan, Italy; giulia.marchetti@unimi.it (G.M.); antonella.darminio@unimi.it (A.d.M.); 5Icona Foundation, 20145 Milan, Italy; alessandro.tavelli@fondazioneicona.org; 6Department of Infectious Diseases, La Sapienza University, Polo Pontino, 04100 Latina, Italy; miriam.lichtner@uniroma1.it; 7Institute of Clinical Infectious Diseases, Catholic University of the Sacred Heart, Policlinico Agostino Gemelli IRCCS, 00168 Rome, Italy; antonella.cingolani@unicatt.it; 8HIV/AIDS Clinical Unit, National Institute for Infectious Diseases Lazzaro Spallanzani IRCCS, 00149 Rome, Italy; stefania.cicalini@inmi.it (S.C.); andrea.antinori@inmi.it (A.A.); 9Division of Infectious Diseases, ASST Grande Ospedale Metropolitano Niguarda, 20162 Milan, Italy; massimo.puoti@ospedaleniguarda.it; 10Centre for Clinical Research, Epidemiology, Modelling and Evaluation (CREME), Institute for Global Health, UCL, London WC1N 1EH, UK; a.cozzi-lepri@ucl.ac.uk

**Keywords:** HIV/HCV coinfected patients, HCV eradication, HCV genotype, lipid metabolism, antiretroviral treatment, darunavir/ritonavir

## Abstract

Objectives: HCV shows complex interactions with lipid metabolism. Our aim was to examine total cholesterol (TC) and low-density lipoprotein cholesterol (LDL-C) changes in HIV/HCV coinfected patients, after achieving sustained virological response (SVR), according to different HCV genotypes and specific antiretroviral use. Methods: HIV/HCV coinfected patients, enrolled in the ICONA and HepaICONA cohorts, who achieved DAA-driven SVR were included. Paired t-tests were used to examine whether the pre- and post-SVR laboratory value variations were significantly different from zero. ANCOVA regression models were employed to estimate the causal effect of SVR and of PI/r use on lipid changes. The interaction between the effect of eradication and HCV genotype was formally tested. Results: six hundred and ninety-nine HIV/HCV coinfected patients were enrolled. After HCV eradication, a significant improvement in liver function occurred, with a significant decrease in AST, ALT, GGT, and total plasmatic bilirubin. TC and LDL-C significantly increased by 21.4 mg/dL and 22.4 mg/dL, respectively (*p* < 0.001), after SVR, whereas there was no evidence for a change in HDL-C (*p* = 0.45) and triglycerides (*p* = 0.49). Notably, the TC and LDL-C increase was higher for participants who were receiving darunavir/ritonavir, and the TC showed a more pronounced increase among HCV genotype 3 patients (interaction-*p* value = 0.002). Conclusions: complex and rapid changes in TC and LDL-C levels, modulated by HCV genotype and PI/r-based ART combinations, occurred in HIV/HCV coinfected patients after SVR. Further studies are needed to evaluate the clinical impact of these changes on the long-term risk of cardiovascular disease.

## 1. Introduction

HCV and host lipid metabolism interact through several different pathways [1]. The virus employs the host’s very low-density lipoproteins (VLDLs) to infect hepatocytes, forming a hybrid particle composed of viral and lipoprotein components sharing the same envelop (lipoviral particle). It is assumed that the association of HCV particles with lipoprotein components provides an efficient escape mechanism from neutralizing antibodies [2,3]. Furthermore, HCV modulates lipid homeostasis by increasing hepatic lipogenesis and concomitantly reducing lipid oxidation and export from hepatocytes [4,5,6]. Accordingly, among HCV patients, low circulating total cholesterol (TC) and LDL cholesterol (LDL-C) serum levels are common findings. Hepatocellular steatosis, low circulating LDL-C, and hypobetalipoproteinemia are particularly common among HCV genotype 3 (GT3) patients, due to viral-mediated inhibition of the microsomal triglyceride transfer protein [5,6,7].

HCV clearance following interferon-free treatments is associated with a significant and rapid increase in circulating TC and LDL-C in the majority of patients, which persists after the end of treatment (EOT) [8,9]. Indeed, the reduction in the production of lipid droplets in HCV-infected hepatocytes reflects the suppression of the HCV core protein induced by DAA treatment and results in a rebound of circulating TC and LDL-C [10].

HIV infected patients are prone to develop atherosclerosis, thus cardiovascular disease (CVD) is one of the most important causes of non-AIDS related morbidity and mortality in this patient population [11,12,13].

Although the CVD risk increase relies on traditional risk factors such hypertension, dyslipidemia, and diabetes, these factors are also promoted and exacerbated by the infection through direct and indirect mechanisms. Antiretroviral therapy (ART) also seems to play a pivotal role in the pathogenesis of cardiovascular damage: large prospective cohort studies documented an increased risk of acute myocardial infarction in patients exposed to abacavir (ABC) and first-generation protease inhibitors lopinavir (LPV) and indinavir (IDV) [14,15].

Recent data from the D: A: D multicohort HIV population reported that cumulative exposure to ritonavir-boosted darunavir (DRV/r) was associated with cardiovascular major events; this association persisted after adjustment for the following potentially confounding factors: hypertension, smoking, previous cardiovascular disease, body-mass index, diabetes, CD4 count, and dyslipidemia [16].

The CVD risk is expected to be even higher among HIV subjects coinfected with HCV due to the synergic triggering effect of the two viruses on the inflammatory response, which could promote CVD occurrence.

Osibogun et al. recently conducted a meta-analysis to clarify and quantify the association between HIV/HCV co-infection and CVD risk. Analyzing four cohort studies with a total of 33.723 participants, they concluded that the pooled adjusted hazard ratio for the association between HIV/HCV co-infection and CVD was 1.24 (95% CI 1.07–1.40) compared to HIV mono-infection [17].

Pursuing these lines of evidence, the present study aimed to investigate the possible effect of DAA-driven HCV eradication on lipid metabolism in HIV/HCV coinfected patients according to HCV genotype and the type of concomitant ART regimens used.

## 2. Patients and Methods

The analysis included patients from the HepaICONA and ICONA cohorts. The HepaICONA is a cohort of HIV/HCV coinfected patients, seen for care in Italy since January 2013, who are naïve to DAAs and had detectable HCV-RNA at enrollment. The analysis also included data from HIV/HCV coinfected patients enrolled in the ICONA cohort (the Italian cohort of ART-naïve patients at enrolment, which was started in January 1997) for those that satisfied the inclusion criteria for this analysis (see below) [18].

Ethics committee approval was obtained from all participating centers for both cohorts, and all included patients provided informed written consent to participate. Protocol codes and dates of approval of the study by Institutional Review Boards (or Ethics Committees) of all participants Centers are listed in Appendix A.

Inclusion criteria were: age ≥ 18 years, active HCV infection (HCV-RNA positive), HIV infection treated with ART for at least 6 months, initiation of DAA and achievement of SVR on treatment, as well as availability of at least one pair of biomarkers (see below).

The exclusion criteria were: current statin use, no response or relapse within 6 months from the EOT, and treatment with IFN-containing anti-HCV regimens.

The baseline was defined as the date of initiation of DAA treatment with a detectable HCV-RNA. Socio-demographic and epidemiologic factors (e.g., age, gender, and HCV genotype) were collected at baseline. The following anthropometric, clinical, HCV and HIV-related parameters were also collected at baseline and at follow-up: body mass index, hepatic stiffness as assessed by liver elastography. Biochemistry, HCV and HIV viral load, CD4+ and CD8+ counts, and current ART regimens used.

The following biomarkers, routinely measured as a part of the patients’ daily care on average every 6 months in the cohorts, were included in this analysis: total LDL and HDL cholesterol, triglycerides, blood glucose, AST, ALT, GGT, total bilirubin, creatinine, INR, hemoglobin, platelets, leucocytes, CD4+ and CD8+ count, and plasmatic HIV-RNA. The main focus of this analysis was on cholesterol (total HDL-C and LDL-C) nd triglycerides.

In order to be included in this analysis, participants had to contribute at least one pair of biomarkers (cholesterol and triglycerides for the main analysis). The first pair of biomarkers (T0 and T1) was defined as the two most recent values in a window (−12; 0) months prior to the date of DAA initiation. The second pair (T1 and T2), on the other hand, consisted of the latest value in the pre-treatment window (T1) and the latest value in the window (+4; +12) months after the end of treatment (EOT). Each participant had the option to contribute the first pair, the second pair, or both.

## 3. Statistical Analysis

The median (IQR) of biomarker values were calculated at each of the three time points (T0, T1, and T2). Within each pair, we calculated unadjusted mean differences and a paired t-test was used to verify whether changes were significantly different from zero. This was done for the full set of considered biomarkers. In order to evaluate the effect of the use of specific antiretroviral drugs (e.g., TAF, TDF, abacavir, and RAL, as well as those belonging to the PI/r class) on lipid changes, in the context of DAA-driven HCV eradication, only the data of the post-DAA initiation pair (T1 and T2) were used. Specifically, a patient was defined as having received a PI/r-based regimen if they were exposed to the same drug of this class both at T1 and T2. An analysis of the covariance (ANCOVA) was performed, fitting a separate model for each drug. Thus, for example, to evaluate the possible causal effect of DRV/r on LDL-C changes, an ANCOVA regression model was fitted, which included LDL-C at time T1, the exact time difference between T1 and T2, the previous use of ATV/r and LPV/r, the time from HIV diagnosis, and the current use of DRV/r. The potential confounders included in the model were chosen on the basis of axiomatic knowledge and previously published data. These assumptions were visualized through the aid of directed acyclic graphs (DAG, Appendix A) [19].

According to these assumptions, in order to estimate the causal effect of DRV/r exposure on LDL-C changes, the following confounding backdoor pathways needed to be blocked:

DRV/r ← previous use of LPV/r → LDL-C;DRV/r ← previous use of ATV/r → LDL-C;DRV/r ← time since HIV diagnosis → LDL-C.

Identical models were fitted for other biomarkers (total cholesterol, HDL-C, and triglycerides).

In order to estimate the marginal effect of DAA-driven HCV eradication on lipids, the combined data from both T0 and T1 and T1 and T2 pairs were used. Changes in biomarkers over T0–T1 were used as a measure of the changes while HCV was still replicating in the absence of DAA. These were then compared to the changes that occurred over T1–T2, potentially reflecting the effect of HCV eradication on lipid changes. The population level mean HDL-C values were also calculated at each month in the time window (−12; +12) months of the date of DAA initiation. Through these data, we then fitted an interrupted time series (ITS) ARIMA model, which tested for correlations in the data by first estimating autoregressive parameters to be included in the model and then the final parameters for the markers intercept and slope before and after DAA. The maximum likelihood was used to fit the model. Twelve lags (one for each month before and after DAA initiation) were tested and lag parameters were included in the final model if statistically significant. Lags were entered into the model using backward elimination in order to fit the most parsimonious model. A Durbin-Watson test was used to test for the presence of autocorrelation. The log likelihood for the overall model was also produced in order to assess the overall quality of the model.

To evaluate the causal effect of DAA-driven HCV eradication, an ANCOVA model was fitted. Due to the fact that some participants contributed both T0 and T1 and T1 and T2 pairs, robust standard errors were calculated to account for the correlation between measures coming from the same person. As we had selected people who had eradicated HCV with DAA, and we used the marker variation observed pre-DAA as the non-exposed group, this selection essentially operated as an instrumental variable for the association between DAA and the variation in the marker post-SVR, as reflected in the DAG (Appendix A). Furthermore, approximately half of the study population was included when both DAA-naïve and DAA-exposed, thus partial exchangeability was automatically established for these individuals. Therefore, the unadjusted analysis should provide the causal effect of DAA-driven eradication on marker changes.

However, a number of common causes of DAA initiation with RBV and biomarker changes post-eradication were identified (i.e., age, liver stiffness >12.5 KPa at DAA start and HCV genotype), which were included in the multivariable model to control for confounding (Appendix A). Indeed, as shown by this second DAG, in order to estimate the causal effect of RBV-based, DAA-driven HCV eradication on, for example, LDL-C, the following confounding backdoor pathways needed to be blocked:

RBV-based DAA-driven HCV eradication ←HCV genotype →LDL-C;RBV-based DAA-driven HCV eradication ←Hepatic stiffness value →LDL-C.

We further evaluated the hypothesis that the effect of DAA-driven HCV eradication on lipids might vary by HCV genotype (i.e., that HCV genotype was not a confounder, but instead an effect modifier). In order to test this hypothesis, we included an interaction term in the ANCOVA model. As there was strong evidence for an interaction, especially for total cholesterol, the results were stratified by HCV genotype after controlling for previous PI/r use and hepatic stiffness alone.

## 4. Results

The analysis included 699 ART-treated HIV/HCV coinfected patients who achieved SVR following the initiation of DAA treatment. Table 1 and Table 2 show participants’ characteristics overall and according to their pair contribution to the analysis. Although approximately half of the participants (*n* = 350, 50%) contributed both pairs of values, there were a minority of individuals (*n* = 82, 12%) who only contributed to the pre-DAA initiation phase of the study. The median age was 52 (IQR 48–55) years and 25.3% were female. The median BMI was 23 (IQR: 21–26) kg/m^2^. At time of starting DAA, the majority of patients (91%) were virologically suppressed (HIV viral load < 50 copies/mL) and 63% showed a CD4+ count of >500/mmc (median CD4+ 600 (IQR: 386–830) cells/mmc). On average, patients were diagnosed with HIV infection 23 years (IQR 16–28) before they started DAA treatment.

At the time of DAA initiation, ART backbone included tenofovir disoproxil-fumarate (TDF) in 49.1%, tenofovir alafenamide (TAF) in 2.1%, and ABC in 20.9%; the third drug was DRV/r in 22.9%, LPV/r in 4.3%, ritonavir-boosted atazanavir (ATV/r) in 16.9%, and raltegravir (RAL) in 19.9% of enrolled subjects.

The median value of hepatic stiffness at DAA initiation, as assessed by FibroScan™, was 10.5 kPa (IQR: 7.2–16.6), and 212 patients (30.3%) had a diagnosis of liver cirrhosis as defined through clinical criteria or hepatic stiffness of >13 kPa. None of the enrolled patients had decompensated liver cirrhosis, which was defined as the presence of ascites, hepatic encephalopathy, hepatorenal syndrome, or esophageal variceal bleeding. Six patients (0.9%) were diagnosed with hepatocellular carcinoma (HCC). The most frequently represented HCV genotype was 1a (40.6%), followed by genotype 3a (HCV GT3a, 22.8%), genotype 4 (13.6%), and genotype 1b (12.5%).

The median baseline pre-DAA treatment ALT and AST levels were slightly over the normality threshold (63, IQR: 38–102 and 51, IQR: 34–84, respectively) and the median basal HCV-RNA was 6.03 (IQR: 5.48–6.48) log10 copies/mL (See Table 2).

Unadjusted pair analyses of the mean changes were conducted for the full set of considered biomarkers (Table 2). Platelet count, ALT, GGT, and HIV viral load showed a decrease in the absence of DAA treatment (T0 and T1 pair), which was significantly different from zero, possibly due to a fluctuant trend with no clinical relevance.

After HCV eradication (T1 and T2 pair), a significant improvement in liver function occurred, with a significant decrease in AST, ALT, GGT, and total plasmatic bilirubin.

On the other hand, TC and LDL-C serum levels, which, prior to DAA, tended to be stable, showed a significant increase after SVR (Δ + 21.4 mg/dL and Δ + 22.4 mg/dL, respectively), whereas high-density lipoprotein cholesterol (HDL-C) and triglycerides appeared to remain almost unchanged (Table 3).

We repeated the analysis after using the latest value in the alternative window (+6; +12) months for T1–T2 and, although having a slightly reduced sample size, the results were similar (see Appendix A).

We also performed the main analysis (Table 3) after removing the 82 patients for which only pre-DAA biomarker measurements were available, and the variations in total cholesterol, LDL-C, HDL-C, and triglycerides over the window of T0–T1, although having a reduced sample size, were similar to those shown in the main analysis (see Appendix A). In this analysis, evidence for a decreasing trend in LDL-C was registered, which was then reversed after initiation of DAA treatment.

The ITS analysis showed statistically not-significant variations in LDL-C during pre- and post-DAA time intervals, with a decrease of 0.05 mg/dL/month *p* = 0.88 and an increase of 0.58 mg/dL/month, *p* = 0.25, respectively. However, in correspondence with the month in which DAA treatment was initiated, a significant increase in LDL-C was estimated (Δ + 12 mg/dL, *p* = 0.002), which was consistent with those shown by the results of the main analysis (Table 3).

HCV eradication had an impact on HIV parameters as well: HIV viral load tended to decrease, whereas CD8+ count showed a significant post-SVR increase (HIV RNA Δ − 0.10 log10 cp/mL, CD8+ Δ + 40.97 cells/mmc, and the mean difference among mean T1–T2 values was *p* < 0.001).

The ANCOVA analysis showed that, after DAA-driven HCV viral clearance, use of DRV/r over the period encompassing HCV eradication was associated with a significant increase in TC, LDL-C, and triglycerides (Δ + 14.2 mg/dL *p* < 0.001, Δ + 13.37 mg/dL *p* = 0.003, and Δ + 21.41 mg/dL *p* = 0.009, respectively), which persisted after adjustment for the length of time since HIV diagnosis and the previous use of other PI/rs. In contrast, the use of LPV/r was associated with an isolated post-SVR increase in TC (Δ + 33.82 mg/dL, *p* = 0.001). After adjustment for length of time since HIV diagnosis and use of PI/r, ABC-exposed patients showed a significant increase in TC and triglycerides levels after DAA treatment (Δ + 9.66 mg/dL, *p* = 0.004 and Δ + 20.45 mg/dL, *p* = 0.008, respectively). TDF-exposed patients showed a significant decrease in TC levels (Δ − 8.43 mg/dL, *p* = 0.005) and an increase in HDL-C levels (Δ + 2.78 mg/dL, *p* = 0.035) after controlling for potential confounders as described in the DAG (Appendix A). Conversely, none of the other associations between specific antiretrovirals (atazanavir, TAF, and raltegravir) and changes in biomarkers were statistically significant (Table 4).

Moreover, we observed that use of sofosbuvir-containing or sofosbuvir-free DAA regimens had a similar impact on SVR-driven lipidic metabolism changes (data not shown).

In addition, the effect of HCV eradication on lipidic metabolism varied according to the participants’ HCV genotype. The increase in the post-SVR TC, LDL-C, and HDL-C increase (Δ + 37.74 mg/dL, Δ + 45.35 mg/dL, and Δ + 2.82 mg/dL, respectively) was almost two-fold higher in HCV GT3a patients versus participants carrying other HCV genotypes (Δ + 21.63 mg/dL, Δ + 29.23 mg/dL, and Δ − 1.29 mg/dL, respectively, in non-GT3a patients after adjustment for time between measurements; see Table 5). The *p*-value for the interaction was only strongly significant for total cholesterol (*p* = 0.002), but large differences by genotype were observed for LDL-C as well.

## 5. Discussion

In this analysis we evaluated the impact of DAA-driven HCV eradication on lipid metabolism in the HIV/HCV coinfected patients from the ICONA and HepaICONA cohort by examining clinical parameters 0–12 month before and 4–12 months after DAA initiation. A major strength of our analysis is that, to the best of our knowledge, it is the first work to examine, in a large study population, how specific HIV drugs might influence lipid metabolism in the setting of HCV eradication.

We found that TC and LDL-C serum levels, which prior to DAA tended to be stable, showed a significant increase after SVR, whereas HDL-C and triglycerides appeared to remain almost unchanged. These findings are consistent with those of previous observations conducted in mono-infected HCV patients also showing a similar lipidic metabolism improvement after HCV eradication [8,9,10,20].

In detail, Meissener et al. demonstrated that post-SVR LDL-C increased only in patients who achieved SVR and that LDL-C declined back to pre-treatment levels in patients who experienced treatment relapse. Furthermore, these authors also noted a correlation between a sharp increase in serum LDL-C and plasmatic HCV-RNA decline, suggesting a direct effect of HCV clearance on serum cholesterol [20]. Unfortunately, the frequency of HCV-RNA measurements and biomarkers in our cohort did not allow for further analysis of the dynamics of this association. To date, data from HIV/HCV coinfected patients are limited and largely come from studies with small sample sizes. Mauss et al. analyzed HIV/HCV coinfected patients from a prospective German multi-center cohort of patients treated with different DAAs (GECCO cohort) and observed a statistically significant on-treatment increase in TC and LDL-C values, which persisted after the EOT. Conversely, HDL-C and triglycerides remained unchanged after HCV eradication [21].

Recently, Townsend et al. also reported a rapid increase in both TC and LDL-C values that was sustained during and after treatment in HIV/HCV coinfected patients treated with ledipasvir (LDV) plus SOF combination therapy for 12 weeks [22]. These findings from a similar study, conducted in a different European setting, are consistent with those observed by in our study, but on a larger scale.

An important finding of our analysis is the impact of HCV genotype on post-SVR metabolic changes. In fact, among the 157 patients with HCV GT3a infection (22.8% of study population), the increase of both TC and LDL-C after DAA treatment was almost two-fold higher than that observed in participants carrying other HCV genotypes (Table 5). This result has biological support as HCV GT3a is known to be associated with lower LDL-C, hypobetalipoproteinemia, and increased liver steatosis due to viral-mediated inhibition of microsomal triglyceride transfer protein [5,6,7]. The majority of other studies conducted to evaluate the post-SVR changes in lipid metabolism were unable to unearth this finding as they enrolled exclusively or mainly non-GT3 patients, both in the HCV mono-infected [8,9,10,20,23,24,25,26] and in the HIV/HCV coinfected population [22].

Younossi et al. examined HCV GT2 and GT3 patients treated with sofosbuvir plus ribavirin to assess changes in the serum lipid and distal (post-squalene) cholesterol biosynthesis metabolite profile and showed that, at week 12 of treatment, HCV GT3 patients had a significantly higher increase of TC, LDL-C, HDL-C, key distal sterol metabolites, and apoB, along with a significant decrease of apoE, when compared to GT2 patients (difference between genotypes: all *p* < 0.05) [27]. Similarly, by week 4 of follow up, GT3 patients showed a significantly higher increase of TC, LDL-C, HDL-C, and apoB compared to GT2 patients. These findings suggest that in GT3 patients, who present a reduced level of circulating lipids, HCV suppression due to antiviral treatment is capable of restoring distal sterol metabolites, removing viral interference with de novo lipogenesis or selective retention by hepatocytes [27]. Our data, for the first time, confirm these results in HIV-positive patients with HCV GT3 co-infection, indicating that the lipid metabolism mechanisms in the HCV/HIV co-infected population do not differ from the HCV mono-infected patients’ population.

Of interest, and conflicting with these and our findings, Mauss et al., in a post-hoc analysis of 327 patients with HCV mono-infection and 193 patients with HCV/HIV coinfection treated with DAAs, found that HCV genotype or the type of HCV treatment regimen had no significant impact on changes of total cholesterol or triglycerides between baseline and post-treatment week 12 [21]. It should be underlined that GT3 represented only 16% of their study population, therefore the analysis might have lacked the power to detect this association.

In parallel with TC and LDL-C levels increase, it was reported that hepatic steatosis, assessed by a controlled attenuation parameter (CAP), seemed to significantly decrease following HCV eradication [26]. In GT3, the induction of liver steatosis is known to be mediated by the HCV core protein that inhibits microsomal triglyceride transfer protein activity and very low-density lipoprotein secretion [28]. Therefore, the disappearance of the core protein may account for the molecular mechanism underlying the improvement in liver steatosis. Unfortunately, the changes in liver steatosis in HCV GT3 patients have not been assessed so far, while several studies have evaluated such changes following antiviral treatment in patients with other genotypes [26,29,30,31,32]. Indeed, an interesting perspective of the present study was to encourage the evaluation of the steatosis evolution in HCV genotype 3 patients, both mono-infected and HIV coinfected, given the role that steatosis plays in the risk of liver disease progression and of HCC development [33,34].

HCV infection was hypothesized as a contributor of poor CD4^+^ recovery in patients living with HIV. In our analysis, HCV eradication did not significantly modify CD4 cell count, whereas an increase in CD8 cell count was documented. A recent study from Bandera et al. similarly described a significant CD8^+^ cell increase in HIV/HCV patients treated with ribavirin (RBV)-containing DAA [35].

To the best of our knowledge, our analysis is the first to investigate the possible role of specific ART regimens on SVR-driven lipid changes in HIV/HCV coinfected patients pre- and post-eradication of HCV. In our cohort of 699 HIV/HCV coinfected patients who achieved SVR following a DAA treatment, we observed that only DRV/r-exposed patients experienced a post-SVR increase in both TC and LDL-C, which persisted after adjustment for the length of time since HIV diagnosis and previous use of PI/r (see Table 4, panel A and B). At the same time, only TDF (but not TAF) exposed patients experienced an increase in HDL-C levels after HCV viral clearance (see Table 4, panel C). Furthermore, DRV/r- and ABC-exposed patients showed a significant increase in triglycerides levels after HCV eradication, whereas, conversely, TDF (but not TAF) seemed to play a protective role.

Interestingly, Ryom et al., in analyzing data from the HIV D: A: D cohort, showed that cumulative exposure to DRV/r was associated with a higher risk of developing major cardiovascular events; dyslipidemia was treated as a potential confounder in this analysis so it is not possible to exclude that this increase in risk was mediated by variations in lipidic metabolism [16].

Of note, ABC-exposed patients also tended to show an increase in triglycerides plasmatic levels after DAA treatment, although it was not significant (Δ + 20.45 mg/dL, *p* = 0.008) (See Table 4, panel D), confirming a possible unfavorable impact of this agent on lipidic metabolism [36].

The impact of post-SVR cholesterol increase on cardiovascular risk is not clearly established yet. However, in HCV mono-infected patients who achieved a DAA-driven SVR, Ichikawa et al. recently observed a one-year significant increase in carotid intima-media thickness (IMT), a highly reproducible, non-invasive, and reliable marker of atherosclerotic process [24].

Before drawing final conclusions, a number of limitations need to be mentioned. First, although careful consideration was given while evaluating all possible confounding factors for the exposure of interest (our assumptions are transparently visualized with the aid of DAGs), we cannot be sure that such assumptions are correct and we cannot rule out the possibility of unaccounted, unmeasured confounding. We are aware that our study population was selected on the basis of having achieved SVR, and, therefore, the potential confounders selected on the basis of being causes of DAA initiation might in fact not be key confounding factors. Reassuringly, however, the results of the unadjusted and adjusted analyses were very similar.

A further important limitation of the study is that not all patients contributed marker pairs to the pre- and post-DAA periods, and there were missing data for some biomarkers. In particular, LDL-C and BMI were only available for about 40% of the study participants. This limited the statistical power, and the results might have been affected by selection bias.

Unfortunately, we were not able to quantify a possible post-SVR increase in CVD risks induced by metabolism changes, because the data necessary to calculate such CVD scores (e.g., systolic arterial blood pressure) was missing for a large proportion of the study population. Indeed, although it was not a key objective, this is a limitation of our analysis and the clinical significance of hypolipidemia reversal among SVR patients, such as the risk of myocardial infarction or cerebral vascular disease, deserves further investigation.

Furthermore, it would have been interesting to also evaluate the controlled attenuation parameter (CAP), which would have added a significant contribution to the analysis. Unfortunately, most of the clinical centers enrolling participants into our study did not routinely perform this exam.

In conclusion, our findings indicated that, along with a significant improvement in liver function, patients undergoing successful DAA-driven HCV eradication should have their serum lipid levels monitored carefully, especially in patients treated with DRV/r or LPV/r, as HCV clearance may worsen the CVD risk of some patients with previously unappreciated coronary risk [37,38]. Further studies are needed to evaluate the clinical impact of lipid changes on the long-term risk of cardiovascular disease in the HIV/HCV coinfected population.

## Figures and Tables

**Table 1 viruses-13-01402-t001:** Participants’ characteristics according to pair contributions. BMI: body mass index; ART: antiretroviral therapy; HCC: hepatocellular carcinoma; TDF: tenofovir disoproxil-fumarate; TAF: tenofovir alafenamide; ABC: abacavir; LPV/r: ritonavir boosted lopinavir; DRV/r: ritonavir-boosted darunavir; ATV/r: ritonavir-boosted atazanavir; and RAL: raltegravir.

Patient Characteristics	Pair Contributions			
	T0–T1 (*n* = 82)	T1–T2 (*n* = 267)	Both (*n* = 350)	Total (*n* = 699)
Age, years, median (IQR)	52 (49–55)	51 (48–54)	52 (49–55)	52 (48–55)
Female sex n (%)	19 (23.2%)	76 (28.5%)	82 (23.4%)	177 (25.3%)
BMI, kg/m2, median (IQR)	22 (19–24)	24 (20–27)	23 (21–26)	23 (21–26)
Time from HIV diagnosis, years, median (IQR)	25 (15–29)	24 (18–29)	21 (15–28)	23 (16–28)
Time since first ART, years, median (IQR)	18 (8–20)	18 (13–20)	17 (10–20)	17 (10–20)
Liver stiffness, kPa, median (IQR)	8.6 (6.1–12.1)	11.2 (7.9–18.6)	10.8 (7.3–16.6)	10.5 (7.2–16.6)
Liver stiffness at DAA > 13 kPa, median (IQR)	16 (19.5%)	89(33.1%)	107 (30.6%)	212 (30.3%)
Decompensated cirrhosis, n (%), median (IQR)	0 (0.0%)	0 (0.0%)	0 (0.0%)	0 (0.0%)
HCC, n (%)	1 (1.2%)	1 (0.4%)	4 (1.1%)	6 (0.9%)
Use of TDF, n (%)	45 (54.9%)	122 (45.7%)	176 (50.3%)	343 (49.1%)
Use of TAF, n (%)	1 (1.2%)	11 (4.1%)	3 (0.9%)	15 (2.1%)
Use of ABC, n (%)	11 (13.4%)	61 (22.8%)	74 (21.1%)	146 (20.9%)
Use of LPV/r, n (%)	2 (2.4%)	10 (3.7%)	18 (5.1%)	30 (4.3%)
Use of ATV/r, n (%)	12 (14.6%)	35 (13.1%)	71 (20.3%)	118 (16.9%)
Use of DRV/r, n (%)	22 (26.8%)	61 (22.8%)	77 (22.0%)	160 (22.9%)
Use of RAL, n (%)	15 (18.3%)	63 (23.6%)	61 (17.4%)	139 (19.9%)

**Table 2 viruses-13-01402-t002:** Participants’ analytic data according to pair contributions. LDL: low-density lipoprotein cholesterol; HDL: high-density lipoprotein cholesterol; GGT: gamma-glutamyl transferase; and INR: international normalized ratio.

Analytical Data	Pair Contributions			
	T0-T1 (*n* = 82)	T1-T2 (*n* = 267)	Both (*n* = 350)	Total (*n* = 699)
HCV viral load at DAA, log10 copies/mL, median (IQR)	6.08 (5.33–6.48)	5.99 (5.49–6.48)	6.05 (5.47–6.48)	6.03 (5.48–6.48)
Blood glucose, mg/dL, median (IQR)	88 (82–102)	90 (82–101)	91 (82–104)	90 (82–102)
Creatinine, mg/dL, median (IQR)	0.63 (0.10–0.83)	0.74 (0.11–0.89)	0.76 (0.51–0.96)	0.75 (0.11–0.92)
Total cholesterol, mg/dL, median (IQR)	162 (139–188)	160 (136–189)	160 (133–188)	160 (135–189)
LDL cholesterol, mg/dL, median (IQR)	92 (74–122)	92 (69–119)	92 (70–121)	92 (70–120)
HDL cholesterol, mg/dL, median (IQR)	44 (37–53)	44 (36–55)	42 (32–53)	43 (34–54)
Triglycerides median (IQR)	112 (73–147)	106 (85–152)	122 (87–175)	112 (84–162)
ALT, mg/dL, median (IQR)	60 (38–92)	58 (36–99)	67 (39–108)	63 (38–102)
AST, mg/dL, median (IQR)	52 (38–75)	47 (34–83)	55 (35–93)	51 (34–84)
GGT, mg/dL, median (IQR)	71 (40–124)	77 (41–125)	82 (47–144)	78 (42–133)
Total bilirubin, mmol/L, median (IQR)	0.50 (0.15–0.80)	0.44 (0.08–0.80)	0.58 (0.22–0.99)	0.51 (0.13–0.89)
Platelets, mg/dL, median (IQR)	185 (151–227)	160 (114–211)	164 (110–213)	165 (116–213)
INR median (IQR)	0.96 (0.88–1.07)	1.02 (0.13–1.14)	0.97 (0.15–1.08)	0.99 (0.14–1.12)
CD4 count, cells/mmc, median (IQR)	562 (433–898)	601 (369–828)	600 (386–825)	600 (386–830)
CD4 at DAA > 500 cells/mmc, median (IQR)	53 (64.6%)	163 (61.7%)	225 (64.3%)	441 (63.4%)
CD8 count, cells/mmc, median (IQR)	851 (634–1054)	882 (606–1214)	815 (566–1129)	842 (585–1154)
CD4/CD8 median (IQR)	0.81 (0.51–1.06)	0.69 (0.42–1.00)	0.74 (0.50–1.05)	0.73 (0.48–1.04)
HIV viral load < 50 copies/mL, n (%)	72 (88.9%)	237 (93.3%)	309 (89.0%)	618 (90.6%)

**Table 3 viruses-13-01402-t003:** Unadjusted pair analyses for all biomarkers. BMI: body mass index; LDL: low-density lipoprotein cholesterol; HDL: high-density lipoprotein cholesterol; GGT: gamma-glutamyl transferase; and INR: international normalized ratio.

Biomarker	Pairs													
	T0–T1 (Both Pre-DAA Treatment)							T1–T2 (Pre- and Post-DAA Treatment)						
	*N*	*Mean1*	*SD1*	*Mean2*	*SD2*	Δ	*p*-*Value*	*N*	*Mean1*	*SD1*	*Mean2*	*SD2*	Δ	*p*-*Value*
BMI, kg/m^2^	139	26.5	27.2	25	21	−1.4	0.33	237	26.1	23.3	28.4	32	2.3	0.19
Blood glucose, mg/dL	551	96.21	30.04	94.88	26.43	−1.32	0.23	868	95.46	26.68	95.31	26.97	−0.16	0.82
Creatinine, mg/dL	658	0.7	0.5	0.8	4	0.1	0.39	1027	0.8	3.2	0.7	0.7	−0.1	0.39
Total cholesterol, mg/dL	432	162.6	43.1	161.5	42.2	−1.2	0.37	617	162	40.7	183.5	41	21.4	<0.01
LDL cholesterol, mg/dL	170	96.6	38.8	94.6	37.7	−2	0.27	235	91.9	34.3	114.4	32.2	22.4	<0.01
HDL cholesterol, mg/dL	249	44.1	14.9	44.1	14.4	0	0.98	360	46	16.8	46.5	14.3	0.5	0.45
Triglycerides, mg/dL	431	137.2	83	134.6	78.1	−2.6	0.45	617	132.7	75.5	134.9	81.1	2.2	0.49
ALT, mg/dL	667	83.79	78.14	78.24	62.42	−5.56	0.02	1047	79.57	64.04	25.58	18.95	−54	<0.01
AST, mg/dL	614	66.49	49.84	64.08	42.65	−2.41	0.13	989	66.2	46.87	27.31	15.64	−38.9	<0.01
GGT, mg/dL	417	126.6	169.7	109.3	112.2	−17.3	<0.01	650	107.9	116.2	47.01	90.8	−60.9	<0.01
Total bilirubin, mg/dL	641	0.68	0.92	0.66	1.04	−0.03	0.49	987	0.67	1.01	0.56	0.8	−0.11	<0.01
Platelets/mmc	670	173.8	77.7	170.4	70.9	−3.4	0.02	1055	308.3	4552	180	72.9	−128	0.36
INR	98	0.95	0.35	0.98	0.34	0.03	0.55	188	0.95	0.38	1	0.37	0.05	0.18
CD4 count, cells/mmc	731	689.2	706.7	860.6	3807	171.4	0.22	1065	811.8	3206	748.2	824.7	−63.7	0.52
CD8 count, cells/mmc	669	923.9	498.8	926	546.3	2.14	0.87	924	916.5	565.4	957.5	577	40.97	0.01
CD4/CD8 ratio	669	0.79	0.43	0.94	3.02	0.15	0.21	924	1.13	5.21	2.46	48.43	1.33	0.41
HIV RNA log10 cp/mL	658	0.92	1.08	0.78	0.9	−0.15	<0.01	978	0.77	0.89	0.68	0.9	−0.1	<0.01

**Table 4 viruses-13-01402-t004:** Mean changes in biomarkers from fitting an ANCOVA regression model for total cholesterol (panel A), LDL cholesterol (panel B), HDL cholesterol (panel C), and triglycerides (panel D) stratified by antiretroviral drugs. TDF: tenofovir disoproxil-fumarate; and TAF: tenofovir alafenamide. * Adjusted for length of time since HIV diagnosis. A separate model for each drug was performed. Models evaluating a drug included in the PI/r class were further adjusted for use of other drugs in the class (e.g., the DRV/r model was controlled for use of ATV/r and LPV/r).

A				
Patients’ Drug Exposure (n)	Total Cholesterol Pre-/Post-DAA Treatment Analysis			
	Unadjusted Difference in Means		Adjusted * Difference in Means	
	Pre-/Post-DAA Treatment Variation (95% CI)	*p*-Value	Pre-/Post-DAA Treatment Variation (95% CI)	*p*-Value
TDF use (187)	−9.63 (−15.1–−4.14)	<0.001	−8.43 (−14.3–−2.59)	0.005
TAF use (159)	0.59 (−11.5–12.65)	0.923	3.26 (−8.80–15.31)	0.597
Abacavir use (152)	6.05 (−0.35–12.45)	0.064	9.66 (3.11–16.21)	0.004
Darunavir use (156)	12.46 (5.91–19.00)	<0.001	14.20 (7.50–20.90)	<0.001
Lopinavir use (12)	30.10 (11.83–48.37)	0.001	33.82 (15.70–51.94)	<0.001
Atazanavir use (51)	3.19 (−5.56–11.94)	0.475	7.48 (−1.23–16.19)	0.093
Raltegravir use (125)	−4.82 (−11.5–1.85)	0.158	−3.92 (−10.7–2.84)	0.256
**B**				
**Patients’ Drug Exposure (n)**	**LDL Cholesterol Pre-/Post-DAA Treatment Analysis**			
	**Unadjusted Difference in Means**		**Adjusted * Difference in Means**	
	**Pre-/Post-DAA Treatment Variation (95% CI)**	***p*-Value**	**Pre-/Post-DAA Treatment Variation (95% CI)**	***p*-Value**
TDF use (187)	−3.61 (−11.0–3.83)	0.343	−0.93 (−9.26–7.41)	0.827
TAF use (159)	13.32 (−6.35–33.00)	0.186	8.29 (−13.0–29.62)	0.447
Abacavir use (152)	1.59 (−7.24–10.41)	0.725	1.61 (−7.94–11.16)	0.742
Darunavir use (156)	11.93 (3.60–20.26)	0.005	13.37 (4.55–22.18)	0.003
Lopinavir use (12)	9.07 (−10.7–28.80)	0.369	14.01 (−5.74–33.77)	0.166
Atazanavir use (51)	−0.92 (−11.7–9.82)	0.867	3.24 (−7.62–14.11)	0.559
Raltegravir use (125)	−1.36 (−10.5–7.79)	0.771	0.57 (−8.82–9.97)	0.905
**C**				
**Patients’ Drug Exposure (n)**	**HDL Cholesterol Pre-/Post-DAA Treatment Analysis**			
	**Unadjusted Difference in Means**		**Adjusted * Difference in Means**	
	**Pre-/Post-Treatment DAA Variation (95% CI)**	***p*-Value**	**Pre-/Post-Treatment DAA Variation (95% CI)**	***p*-Value**
TDF use (187)	2.39 (0.11–4.67)	0.041	2.78 (0.21–5.35)	0.035
TAF use (159)	−0.56 (−6.45–5.34)	0.853	−2.82 (−8.95–3.31)	0.368
Abacavir use (152)	−1.15 (−3.87–1.56)	0.405	−1.41 (−4.30–1.47)	0.338
Darunavir use (156)	−1.43 (−4.01–1.15)	0.279	−1.06 (−3.75–1.62)	0.437
Lopinavir use (12)	6.03 (−0.53–12.58)	0.072	5.59 (−1.04–12.22)	0.099
Atazanavir use (51)	0.49 (−3.20–4.18)	0.797	0.27 (−3.56–4.10)	0.891
Raltegravir use (125)	1.35 (−1.31–4.02)	0.32	1.45 (−1.35–4.24)	0.311
**D**				
**Patients’ Drug Exposure (n)**	**Triglycerides Pre-/Post-DAA Treatment Analysis**			
	**Unadjusted Difference in Means**		**Adjusted * Difference in Means**	
	**Pre-/Post-DAA Treatment Variation (95% CI)**	***p*-Value**	**Pre-/Post-DAA Treatment Variation (95% CI)**	***p*-Value**
TDF use (187)	−28.6 (−40.5–−16.6)	<0.001	−28.8 (−41.8–−15.8)	<0.001
TAF use (159)	8.06 (−18.7–34.84)	0.556	15.68 (−12.1–43.41)	0.269
Abacavir use (152)	14.21 (0.08–28.34)	0.049	20.45 (5.39–35.50)	0.008
Darunavir use (156)	21.45 (6.80–36.10)	0.004	20.41 (5.14–35.68)	0.009
Lopinavir use (12)	10.62 (−30.5–51.70)	0.612	13.86 (−27.6–55.29)	0.512
Atazanavir use (51)	−2.28 (−21.8–17.20)	0.818	2.67 (−17.4–22.70)	0.794
Raltegravir use (125)	−2.91 (−16.7–10.90)	0.68	−3.81 (−17.9–10.28)	0.597

* adjusted for length of time since HIV diagnosis.

**Table 5 viruses-13-01402-t005:** Panels A, B, C and D. Estimate of the marginal effect of HCV genotype 3 on lipids using pre-DAA and post-DAA pairs. LDL: low-density lipoprotein cholesterol; and HDL: high-density lipoprotein cholesterol. * Adjusted for time between measurements.

Biomarker	Adjusted * Difference in Lipid Changes		
	HCV Genotype 3	Other HCV Genotypes	
	Pre-/Post-DAA Variation (95% CI)	Pre-/Post-DAA Variation (95% CI)	Interaction *p*-Value
Total cholesterol, mg/dL	37.74 (26.13–49.35)	21.63 (16.09–27.18)	0.002
LDL cholesterol, mg/dL	45.35 (26.38–64.33)	29.23 (20.87–37.58)	0.2
HDL cholesterol, mg/dL	2.82 (−3.29–8.94)	−1.29 (−3.82–1.25)	0.17
Triglycerides, mg/dL	−4.17 (−25.5–17.16)	12.65 (−1.20–26.50)	0.64

* adjusted for time between measurements.

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
