# Peer review of "Impact of HCV Eradication on Lipid Metabolism in HIV/HCV Coinfected Patients: Data from ICONA and HepaICONA Foundation Cohort Study"

_viruses, 2021, doi:10.3390/v13071402_

Round 1

Reviewer 1 Report

Dr Spaziante et al in this study examine the impact of HCV eradication on cholesterol and triglycerides In HCV/HIV co-infected patients. Although it has some limitations such as the relatively small number of patients with pre-DAA data (N=82) or lack of patients with decompensated cirrhosis, the paper is well written and the results interesting.

Minor comments

Abstract

The extension of the methods is greater than the results. I suggest to resume the methods and to add further data on the results paragraph.

Material and methods

- Mean (SD) of biomarkers…- The majority of results are provided as Median and IQR.

- Why the use of statins was considered a exclusion criteria? It could be analyzed as a cofactor if the number of patients undergoing statins would be enough.

- It would be interesting to have data on CAP.

- Data on post-DAA CT and TG were considered within the period of +4 and +12 months. Did the results differ when patients with data at +6 month or +12 month post-EOT were considered separately?

Results

- Time of HIV infection is provided, but what was the time on HAART?

- Page 7, line 242- Figure 2?- Where it is?

- Page 7, line 246- Bandera A et al- This reference may be omitted since that article is not published yet, and the results are also showed in the present article.

Tables

- It may be better to turn out table 1 in 2 different tables, one with demographic characteristics and another with analytical data.

- Table 3- Add the number of patients included at each time point, not just the number of each ART regimen. The same for table 4.

Figures

Figure 1- The acronym DAG may be avoided at the figure legend. I think this figure may be moved to the supplementary material.

Author Response

We thank the Editor and the referees for the opportunity to revise our manuscript in light of the concerns raised. Please find attached our point by point responses to their comments.

Reviewer 2 Report

This study contains substantial number of samples, however, I believe the inclusion criteria should be more strict. Including samples with only pre DAA biomarkes in the study that inspects the effect of HCV eradication seems a bit of a stretch. I would strongly advise against including those 82 patients in the study, the number of participants would not decrease much and the study title would be a much better fit.

Author Response

(The authors gave the same response as above.)
